# Prognostic Value of a Multivariate Gut Microbiome Model for Progression from Normal Cognition to Mild Cognitive Impairment Within 4 Years

**DOI:** 10.3390/ijms26104735

**Published:** 2025-05-15

**Authors:** Anne Bauch, Julia Baur, Iris Honold, Matthias Willmann, Greta Louise Weber, Stephan Müller, Sebastian Sodenkamp, Silke Peter, Ulrich Schoppmeier, Christoph Laske

**Affiliations:** 1Department of Psychiatry and Psychotherapy, University of Tübingen, 72076 Tübingen, Germany; iris.honold@med.uni-tuebingen.de (I.H.); greta-louise.weber@student.uni-tuebingen.de (G.L.W.); stephan.mueller@med.uni-tuebingen.de (S.M.); sebastian.sodenkamp@med.uni-tuebingen.de (S.S.); christoph.laske@med.uni-tuebingen.de (C.L.); 2SYNLAB MVZ Leinfelden-Echterdingen GmbH, Labor Dr. Bayer, 70771 Leinfelden-Echterdingen, Germany; will80@gmx.de; 3German Center for Neurodegenerative Diseases (DZNE), 72076 Tübingen, Germany; 4Institute of Medical Microbiology and Hygiene, University of Tübingen, 72076 Tübingen, Germany; silke.peter@med.uni-tuebingen.de (S.P.); ulrich.schoppmeier@med.uni-tuebingen.de (U.S.); 5Section for Dementia Research, Hertie Institute for Clinical Brain Research, Department of Psychiatry and Psychotherapy, University of Tübingen, 72076 Tübingen, Germany

**Keywords:** Alzheimer’s disease, mild cognitive impairment, longitudinal observational study, prediction model, gut microbiome

## Abstract

Little is known about the dysbiosis of the gut microbiome in patients with mild cognitive impairment (MCI) potentially at risk for the development of Alzheimer’s disease (AD). So far, only cross-sectional differences and not longitudinal changes and their prognostic significance have been in the scope of research in MCI. Therefore, we investigated the ability of longitudinal taxonomic and functional gut microbiome data from 100 healthy controls (HC) to predict the progression from normal cognition to MCI over a 4-year follow-up period (4yFU). Logistic regression models were built with baseline features that best discriminated between the two groups using an ANOVA-type statistical analysis. The best model for the discrimination of MCI converters was based on functional data using Gene Ontology (GO), which included 14 features. This model achieved an area under the receiver operating characteristic curve (AUROC) of 0.84 at baseline, 0.78 at the 1-year follow-up (1yFU), and 0.75 at 4yFU. This functional model outperformed the taxonomic model, which included 38 genera features, in terms of descriptive performance and showed comparable efficacy to combined analyses integrating functional, taxonomic, and clinical characteristics. Thus, gut microbiome algorithms have the potential to predict MCI conversion in HCs over a 4-year period, offering a promising innovative supplement for early AD identification.

## 1. Introduction

Mild cognitive impairment (MCI) characterizes a stage of cognitive functioning with a measurable decline in domains such as memory, attention, or language that exceeds normal age-related loss, but does not impact daily living functioning [1]. Although the pathologic etiology of MCI is heterogenous, one potential cause of these symptoms relates to degenerative changes in the brain. About 5–15% of patients with MCI develop Alzheimer’s disease (AD) per year. Therefore, MCI is often considered a precursor stage of developing dementia, such as AD [2].

AD, as the most common form of dementia in elderly patients, is a neurodegenerative disorder of the brain. The pathology is characterized by significant cognitive decline in various memory functions as well as other cognitive domains, including linguistic abilities, visuo-constructive skills, and executive functions. Furthermore, the disease is marked by a substantial loss of everyday competencies and increasing dependency on intensive care. Thus, the early identification of underlying AD pathology is crucial for initiating appropriate therapeutic interventions. In clinical practice, this is typically achieved through a combination of clinical, neuropsychological, biochemical, and imaging-based diagnostic approaches.

Recent research findings suggest that the composition of the gut microbiome may also provide valuable diagnostic insights and be associated with the pathogenesis of neurodegenerative disorders, such as AD pathology or MCI. The human microbiome consists of a complex community of trillions of micro-organisms, including bacteria, archaea, fungi, and viruses, with approximately 95% residing in the gastrointestinal tract. Beyond their fundamental roles in metabolic processes such as digestion and nutrient absorption, these micro-organisms play a critical role in defending against pathogenic agents and maintaining overall health [3]. The gut–brain axis (GBA), a bidirectional communication system connecting the gastrointestinal tract and the central nervous system, has emerged as a significant focus of research. This axis utilizes multiple signaling mechanisms, including the vagus nerve, the enteric and autonomic nervous systems, the immune system, and biochemical mediators such as neurotransmitters and microbial metabolites (e.g., short-chain fatty acids [SCFAs] and branched-chain amino acids [BCAAs]) [4].

These pathways enable bidirectional information transfer, potentially impacting both brain and gut functions, as well as overall health. Emerging evidence suggests that dysbiosis—imbalances in the gut microbiota composition, and diversity—may contribute to the pathogenesis of MCI, AD, and other neurodegenerative and psychiatric disorders [5,6]. Contributing factors include alterations in microbiota distribution, intestinal barrier dysfunction, and inflammatory changes in the intestinal epithelium, commonly referred to as a “leaky gut” [5,6].

Altered gut microbiome compositions are well-documented in cross-sectional studies in patients with AD [7,8,9,10,11], and even in patients with preclinical AD [12]. Furthermore, findings from recent meta-analyses confirm gut microbiological abnormalities in MCI and highlight their role as potential biomarkers for the early identification and diagnosis of AD. However, to assess the prognostic validity of the gut microbiome composition, longitudinal studies are needed. In a recent longitudinal study of our working group [13], we evaluated the prognostic validity of taxonomic and functional gut microbiome models to predict the conversion from MCI to AD over a time frame of 4 years. Taxonomic models (including 24 genera) and functional models (including 25 Gene Ontology [GO] features and 33 Kyoto Encyclopedia of Genes and Genomes ortholog [KO] features) each demonstrated superior prognostic value compared to a clinical-only model (which included age, gender, body mass index [BMI], and Apolipoprotein E [ApoE] genotype), highlighting the importance of incorporating gut microbiome data to predict AD conversion. This raises the question of whether the development of such prediction models could also be informative at an earlier stage of disease. Therefore, our main research question was to identify the most stable multivariate model predicting the progression from normal cognition to MCI based on gut microbiome composition.

Our primary aim was to exploratorily investigate how taxonomic and functional gut microbiome profiles, in conjunction with clinical parameters, can serve as prognostic markers for the progression from healthy controls (HCs) to MCI. By assessing the predictive power of these microbiome data over a 4-year follow-up (4yFU), this study may provide insights into the potential mechanistic links between the gut microbiota, their metabolites, and cognitive decline.

## 2. Results

### 2.1. Patients Demographics

Of the 100 healthy elderly participants included at baseline, 14 dropped out of the study during 4yFU. Only the participants who completed the study were included in the analyses (*n* = 86). Based on clinical classification, *n* = 29 of these individuals developed MCI during the 4yFU, and *n* = 57 remained cognitively stable HCs.

The clinical and demographic characteristics (Table 1) showed no significant differences between HCs and MCI-converted participants at baseline (*p*s > 0.05). As expected, the groups differed significantly at MMSE score at 1yFU and 4yFU, with MCI converters scoring significantly lower than HCs, respectively (*p*s ≤ 0.04).

### 2.2. Discriminatory Ability of the Gut Microbiome Between Stable Healthy Controls and MCI Converters

Logistic regression analyses were used to investigate the discriminatory potential of the gut microbiome to predict the conversion from HCs to MCI using ROC analyses.

**Taxonomic model:** The taxonomic model including 38 genera (Table 2) yielded an AUROC of 0.72 at BL, 0.70 at 1yFU, and 0.58 at 4yFU. We found that 11 of the identified taxa (28.9%) belonged to the phylum Firmicutes, 17 (44.7%) to the phylum Pseudomonadota, 3 each (7.9%) to the phyla Actinomycetota and Euryarchaeota, and 1 taxon (2.6%) each to the remaining phyla (Bacteroidota, Fusobacteria, Elusimicrobiota, and Chloroflexota). The genus Merdimonas, which belongs to the phylum Firmicutes, was significantly increased (*p* < 0.05) in converters. Significantly decreased levels (*p*s < 0.05) in converters were found for the genus Methanobrevibacter (Euryarchaeota) and the genus Thermovenabulum (phylum Firmicutes).

**Functional model:** The functional model containing 14 Gene Ontology (GO) features yielded an AUROC of 0.84 at BL, 0.78 at 1yFU, and 0.75 at 4yFU (see Figure 1). The 14 genera included in this model are listed in Table 3. Significantly elevated levels (*p*s < 0.01) in MCI converters compared to HCs were found for the features GO.0015416 (ABC-type phosphonate transporter activity) and GO.0051484 (isopentenyl diphosphate). Significantly increased levels (*p*s < 0.05) developed in MCI converters for GO.0004339 (glucan 1,4-alpha-glucosidase activity), GO.0043802 (hydrogenobyrinic acid a,c-diamide synthase), and GO.0060567 (the negative regulation of termination of DNA-templated transcription). A significantly decreased abundance (*p* < 0.05) was detected for the feature GO.0004491 (methylmalonate-semialdehyde dehydrogenase), and a strongly decreased abundance (*p* < 0.01) for GO.0015667 (DNA methyltransferase).

**Clinical model**: As clinical characteristics like age, gender, BMI, and ApoE4 status are well-established risk factors for late-onset AD [14,15] and also influence the gut microbiome composition [16,17,18], the prognostic value of a model only including these clinical characteristics was also investigated. This model yielded an area under the receiver operating characteristic curve (AUROC) of 0.57 at BL, 0.54 at 1yFU, and 0.55 at 4yFU (see Table 4).

**Combined analysis:** Using an ensemble learning model including the above mentioned taxonomic, functional, and clinical features yielded an AUROC of 0.84 at BL, 0.78 at 1yFU, and 0.75 at 4yFU (see Table 4).

## 3. Discussion

In this study, we analyzed the prognostic value of the gut microbiome in predicting the conversion from healthy individuals to MCI over a four-year period. In our analyses, we used the taxonomic and functional profiling of the gut microbiome, along with well-established clinical characteristics (BMI, age, ApoE4 status, and gender) to identify the model with the most temporally stable features. We identified a taxonomic model with 38 features and a functional model based on GO with 14 features, both of which were able to accurately predicted MCI conversion after four years. Our findings suggest that changes in bacterial taxa at the community level rather than individual alterations may be associated with the development of MCI. Furthermore, both microbiome models descriptively outperformed the clinical model with the four well-established clinical features (BMI, age, gender, and ApoE4 status), thus highlighting the utility of gut microbiome data in predicting cognitive decline over time.

Looking at both gut microbiome models more specifically, the functional model (using GO features) proved to be more accurate and stable over time in identifying MCI converters with an AUROC of 0.84 at BL, 0.78 at 1yFU, and 0.75 at 4yFU, compared to the taxonomic model (using genera features) with an AUROC of 0.72 at BL, 0.70 at 1yFU, and 0.58 at 4yFU. This means, for example, that persons with MCI could be correctly identified with an accuracy of 75% after four years by using the GO model trained with microbiome data from the baseline.

Furthermore, the combined model (taxonomic, functional, and clinical features) did not achieve a higher accuracy than the GO model alone, implying that the effect of the combined analysis seems to be driven by functional gut microbiome data. The relative stability of the identified GO model over time—with a decline from 84% at baseline to 75% at 4yFU—should be highlighted as a strength of the present study. From a clinical perspective, it might be useful to focus on a limited but time-stable set of features, as these results could be more easily used for diagnostic and therapeutic interventions at an earlier stage of disease development. Such stable gut microbiome patterns have also already been identified in our research group predicting the conversion from MCI to AD [13], but also in other diseases, e.g., Parkinson disease [19].

Furthermore, and in comparison to our previous published work [13], the prognostic power of the best-fitting model in this study (GO model) was still good but descriptively lower than in our previous study. Additionally, the prognostic power slightly decreased over time in the present study and the identified taxonomic and functional features differed from the identified features in our previous study. A potential explanation for these findings might be attributed to the heterogeneity of MCI pathology in comparison to AD pathology. Given the diverse etiological origins of MCI [20], which, not consequently, leads to AD, but may also be identified in patients with other psychiatric disorders or other types of dementia [21], it is plausible that gut microbiomes influence disease progression through distinct pathways. More specifically, in our studies, we used completely different study samples; thus, the cohort of MCI converters in the present study did not correspond to the population of MCI patients who progressed to AD in the previous study. However, as different features have been identified, a model predicting the conversion from HCs to AD would be valuable.

Focusing on the identified microbiome features, our most effective analytical approach, the GO model, revealed a significant upregulation of ABC transporters (ABC-type phosphonate transporter, GO.0015416) and the glycogen-degrading enzyme glucan 1,4-alpha-glucosidase (GO.0004339) in individuals with MCI. Previous evidence indicates that an increased ABC transporter expression plays a contributory role in neurodegenerative processes [22]. Furthermore, an impaired glucose metabolism and reduced neuronal glucose uptake are well-documented in AD [23]. The observed elevation of glucan 1,4-alpha-glucosidase, an enzyme responsible for glycogen hydrolysis and glucose release, may represent a compensatory response to glucose deficiency, a hallmark of AD and potentially a feature of MCI converters.

Two additional features associated with DNA modification merit further discussion. An enzyme associated with the negative regulation of DNA-templated transcription termination (GO.0060567) exhibited increased expression, which may lead to premature transcriptional termination, potentially altering gene expression patterns by modulating transcript diversity and abundance. Given the established role of epigenetic mechanisms in the pathogenesis of MCI and AD [24], this alteration, although nonspecific, may contribute to MCI progression. However, further investigation is necessary in order to elucidate its precise impact.

A second epigenetic regulator displaying dysregulation in MCI was DNA methyltransferase (cytosine-N4-specific), an enzyme critical for DNA methylation during epigenetic modifications. Recent studies have highlighted the pivotal role of epigenetic mechanisms in memory formation and cognitive processes [24]. Hypomethylation, frequently observed in the context of reduced S-adenosylmethionine (SAM) levels, has been associated with diminished folate availability and elevated homocysteine concentrations—both recognized as risk factors for AD [25]. Additionally, DNA hypomethylation has been shown to enhance amyloid beta (Aβ) production through the upregulation of genes involved in plaque formation (APP, PSEN1, and BACE1) [26]. The observed reduction in DNA methyltransferase levels in MCI converters suggests that aberrant epigenetic regulation may contribute to cognitive decline.

We also observed an increase in hydrogenobyrinic acid a,c-diamide synthase (CopB, GO.0043802) in MCI converters. This enzyme participates in the biosynthesis of cobalamin (vitamin B12) and facilitates a reaction that generates glutamate as a byproduct. Excess glutamate accumulation in the brain has been implicated in AD pathogenesis, with studies in transgenic mouse models demonstrating that elevated glutamate levels potentiate Aβ-mediated excitotoxic neuronal activity, a key feature of AD [27]. These findings suggest that excessive glutamate production may exacerbate neurodegenerative processes, potentially accelerating the progression from MCI to AD.

Isopentenyl diphosphate (GO.0051484), an enzyme involved in cholesterol biosynthesis, exhibited increased levels in MCI converters, suggesting an upregulated cholesterol metabolism. Elevated cholesterol levels have been involved in multiple pathways associated with AD pathogenesis and prevalence [28]. Evidence indicates that cholesterol metabolism is dysregulated in AD [29], and Aβ accumulation in the brain appears to be modulated by cholesterol [30]. Although the precise role of cholesterol in MCI and AD progression remains incompletely understood, these findings underscore a potential mechanistic link between cholesterol dysregulation and neurodegenerative processes.

Finally, we identified a significant downregulation of methylmalonate-semialdehyde dehydrogenase (MMSDH; GO.004491), a mitochondrial matrix enzyme involved in the terminal steps of BCAA metabolism, specifically isoleucine, valine, and leucine. A dysregulated BCAA metabolism has been reported in AD, with studies documenting both increased and decreased BCAA levels in affected individuals [31]. The altered expression of MMSDH in MCI converters suggests a potential role in cognitive decline and early AD pathology, highlighting the need for further research to elucidate the mechanistic interplay between BCAA metabolism and neurodegenerative progression.

In the taxonomic genera model, 8 of 17 genera (47.07%) from the Gram-negative phylum *Pseudomonadota* showed an increased abundance in MCI converters. Previous studies have established a correlation between Gram-negative bacteria and AD pathology. Lipopolysaccharides (LPSs), key components of the outer membranes of Gram-negative bacteria, have been suggested as potential pathogenic factors in AD, with elevated levels contributing to disease progression. LPS is associated with exacerbated neuroinflammation and neurodegeneration, as well as increased tau hyperphosphorylation and gut microbiota dysbiosis [32].

Overall, a reduction was observed in 21 of 38 (55%) analyzed bacterial taxa. An altered gut microbiota composition, particularly a decline in bacterial diversity, has been previously linked to neurodegenerative diseases [33] and has been specifically documented in AD [7]. This suggests that gut microbiota dysregulation may also contribute to the onset of MCI in the studied cohort. The three genera that exhibited the most significant alterations in MCI converters were *Thermovenabulum*, *Merdimonas*, and *Methanobrevibacter*.

The genus *Thermovenabulum*, which was significantly reduced in MCI converters, belongs to the phylum *Firmicutes*. *Firmicutes* are known fermentative bacteria involved in the synthesis of SCFAs, organic acids, and gases. Recent research has identified abnormal SCFA concentrations in AD patients, with SCFAs postulated to exert anti-inflammatory effects, inhibit tau protein aggregation, and contribute to the maintenance of blood–brain barrier integrity [34]. Additionally, molecular hydrogen (H_2_), a byproduct of SCFA metabolism, has been proposed as an anti-inflammatory metabolite, with reduced levels being associated with an increased risk of MCI [35]. Given that SCFAs have been shown to exert protective effects against AD pathogenesis, their depletion in MCI converters suggests a potential loss of neuroprotective mechanisms, which may contribute to disease progression.

The genus *Methanobrevibacter*, classified within the phylum *Archaea*, also exhibited a significant reduction in MCI converters. *Methanobrevibacter* species are methanogenic archaea that primarily metabolize CO_2_ and H_2_ and are hypothesized to promote the growth of *Firmicutes* and *Bacillota* bacteria, both of which are key H_2_ producers [36]. According to Hatayama et al., a decline in *Firmicutes* leads to a subsequent reduction in H_2_ production, thereby diminishing anti-inflammatory responses and impairing gut barrier integrity, ultimately facilitating MCI development [35]. If *Methanobrevibacter* depletion leads to a cascade effect, resulting in a further decline in *Firmicutes* and *Bacillota*, it may contribute to the pathophysiological mechanisms underlying MCI. Further investigation is warranted in order to elucidate the precise mechanistic interplay.

*Merdimonas*, the third significantly altered genus, also exhibited a marked decline in MCI converters. This finding aligns with the aforementioned Japanese study, which similarly documented a reduction in *Merdimonas* populations in individuals with MCI [35]. The consistency of this observation across different cohorts suggests that *Merdimonas* depletion may be a relevant microbial signature associated with early cognitive decline, meriting further functional and mechanistic exploration.

Despite the promising results, a few limitations of the study need to be addressed. One of our main limitations is our exploratory study design and the lack of a priori power calculation, which is especially critical concerning the large number of possible features for statistical selection which might result in a serious risk of overfitting. Therefore, larger, more diverse, multi-center studies are needed to validate these findings in an independent sample, thereby increasing statistical power. Additionally, it is important to note that the analyzed models were only compared descriptively rather than statistically, meaning the superiority of the identified model is not definitively established. Furthermore, as we conducted a longitudinal observational study, we had to deal with the challenge of an imbalanced sample (more stable healthy controls compared to MCI converters), which might have influenced our results. Regarding our statistical methods in more detail, ANOVA-type statistics for feature selection might be a valid but rather arbitrary statistical option with several limitations. Therefore, future studies should use more complex, data-driven statistical methods for variable selection (e.g., LASSO and Boruta) instead. Additionally, the use of AUROCs in the statistical analyses does not distinguish between high sensitivity and specificity, which could be important for clinical applications.

Contrary to our expectations, the clinical model including well-established risk factors (age, gender, BMI, and ApoE4) yielded only a poor prognostic value at all time points (AUROC ranging from 0.55 to 0.57), compared to the results of our previous study [13]. This might be attributed to different sample characteristics, as these risk factors might be more relevant in the prediction of AD compared to MCI. This, on the other hand, strengthens our result of the predictive value of the gut microbiome model in this earlier stage of disease. On the other hand, this might also question our statistical approach and the validity of our results and should, therefore, be incorporated in future studies. In comparison to our previous work, the different sample populations (conversion of HC to MCI vs. MCI to AD) may explain the variations in models identified in this analysis. Future studies should also focus on investigating the conversion from HCs to AD. Lastly, it is important to note that the results of our analyses postulate a direct pathway between microbiome changes and cognitive decline. However, there might also be other relevant factors mediating this association, like depression, cardiovascular disease, and metabolic disorders [37].

Our study supports the prognostic value of the functional gut microbiome in predicting the conversion from healthy controls to MCI. A key strength is its longitudinal design, which enhances previous cross-sectional findings [11,12], suggesting a causal link between gut microbiome dysbiosis and cognitive decline. This evidence could inspire future studies focusing on the specific clinical implications of our results (e.g., the identification of specific gut microbiome profile alterations in healthy controls at risk for MCI or potential early interventions targeting the gut microbiome to reduce cognitive decline).

In conclusion, we identified stable gut microbiome algorithms that predict the progression from HCs to MCI over a 4-year follow-up. The present findings highlight the prognostic value of the gut microbiome for the early identification of patients at an increased risk for MCI, which might supplement existing diagnostic assessment methods. To our knowledge, this is the first study investigating gut microbiome features in healthy individuals as prognostic markers for MCI, confirming a link between gut microbiome dysbiosis and cognitive decline. Thus far, previous research established the composition of the gut microbiome as a potential contributing factor to the development of neurodegenerative diseases such as AD. By focusing specifically on the transitional stage of MCI, our research extends the field of neurodegenerative disease studies at a much earlier point in the disease’s progression and complement the results of our previous study. Our findings might be informative for designing future studies focusing on concrete diagnostic tools in clinical practice, having the potential to enhance the early, non-invasive detection of these conditions by using gut microbiome data. Notably, microbiome signatures may be detectable in cognitively asymptomatic individuals, potentially indicating future MCI progression. Especially relevant to clinical practice might be the identification of gut microbiome signatures that are unique to MCI with an underlying AD compared to MCI due to other etiologic causes. Thus, research in that field could, overall, provide a critical time advantage for early diagnosis and targeted therapeutic interventions. Future studies should build on these findings to explore the causal relationship between microbiome dysbiosis and MCI development, in order to improve diagnostic accuracy and supplement existing diagnostic assessment methods, thereby facilitating the development of early interventions for MCI, preclinical dementia, and other neurodegenerative diseases.

## 4. Materials and Methods

### 4.1. Participants

The subjects participating in the AlzBiom study were recruited at the Section of Dementia Research at the Department of Psychiatry and Psychotherapy in Tübingen (Germany) starting from 2016 to end of 2018. All participants were clinically examined at BL by means of routine diagnostic work-up for dementia including physical, neurological, and psychiatric examinations as well as brain imaging. At BL, inclusion criteria were (1) normal cognition assessed via Mini-Mental State Examination ([38]; MMSE ≥ 27) and the Clinical Dementia Rating scale [39,40] CDR = 0), (2) no subjective cognitive decline (SCD), and (3) no history of neurological or psychiatric disorders. For MCI detection, we used the amnestic MCI criteria proposed by Petersen et al. [41]. Furthermore, MMSE and CDR were repeated at 1yFU and 4yFU. Of the 100 participants, 57 participants remained stable over 4 years, 29 participants developed mild cognitive impairment, and 14 dropped out of the study. No a priori power analysis was conducted to assess the sample size. The local ethical committee approved the study and written informed consent was obtained from each individual.

### 4.2. Determination of the ApoE4 Genotype

The procedure for determining the ApoE4 genotype was performed as previously described [8,13]. DNA was isolated from the cellular fraction of the blood using a proteinase K digestion and subsequent alcohol precipitation. APOE genotyping was performed using the Applied Biosystems Assay-on-demand TaqMan^®^ SNP (Thermo Fisher Scientific, Pittsburgh, PA, USA) Genotyping Assays C_3084793_20 and C_904973_10, which correspond to the APOE SNPs rs429358 and rs7412, and was carried out on a StepOne Real-Time PCR Systems instrument (Thermo Fisher Scientific, Pittsburgh, PA, USA). The ApoE ε4-positive genotype was assigned if at least one ε4 allele was identified.

### 4.3. Stool Collection, DNA Extraction, and Shotgun Metagenome Sequencing

As previously described [8,13], the stool samples were collected in a sterile plastic device (Commode Specimen Collection System, Thermo Fisher Scientific, Pittsburgh, PA, USA) and were usually collected by the participants at home using the DNA/RNA Shield Fecal Collection Tube R1101 (Zymo Research, Irvine, CA, USA) and sent to our laboratory immediately. In our laboratory, the samples were stored at −20 °C and DNA extraction was performed on the same day with the ZymoBiomics DNA Miniprep Kit D4300 (Zymo Research, Irvine, CA, USA). At GATC Biotech AG (Constance, Germany), DNA extraction was performed in batches of 12 to 18 samples at the end of the sample collection period. Shotgun metagenome sequencing was then conducted at the same facility using the NEBNext Ultra DNA Library Kit (New England Biolabs, Ipswich, MA, USA) for DNA library preparation. Sequencing was carried out on an Illumina HiSeq platform (Illumina, San Diego, CA, USA), employing a paired-end sequencing approach with a targeted read length of 150 bp and an insert size of 550 bp. The objective was to achieve an average sequencing depth of 40 to 50 million reads per sample.

### 4.4. Metagenomic Assembly

As previously described [8,13], we used Trimmomatic (version 0.35) to acquire high-quality reads by means of adapter removing and a sliding window trimming with a minimum length of 100 bp [42]. For quality control of trimmed reads, we used FastQC version 0.11.5 (https://www.bioinformatics.babraham.ac.uk/projects/fastqc/, accessed on 6 March 2023). Metagenomic scaffolds were assembled using SPAdes (version 3.9.0) with a minimum length of 1000 bp to ensure high-quality profiling [43].

### 4.5. Taxonomic Classification

Kraken [44] was used for host removal and taxonomic profiling was performed with MetaPhlAn (Metagenomic Phylogenetic Analysis) [45]. After collection, read counts of input samples observed at taxa levels were normalized using the rarefy function implemented in the vegan bioconductor package (version 2.6-4) [46] in order to compare species richness.

### 4.6. Functional Classification

Functional profiles were analyzed using HUMAnN 2.0 (the HMP unified Metabolic Analysis Network; version 0.11.2) [47]. In accordance with OUT and PICRUSt (Phylogenetic Investigation of Communities by Reconstruction of Unobserved States) [48], the functional categories were identified based on a comparison of the Gene Ontology (GO) Resource (http://geneontology.org/, accessed on 6 March 2023).

### 4.7. Statistical Analysis

We used the statistical software package SPSS (version 23) to analyze demographic and clinical data. For all tests, significance level was set at a priori for α = 0.05. Levene’s test was used to prove the homogeneity of variances. For continuous variables (BMI, age, etc.), *t*-tests for independent samples were used in case of continuous variables. If assumption of normal distribution was not met, we used the nonparametric Mann–Whitney U-test (e.g., MMSE and GDS). For categorical variables (e.g., gender distribution and ApoE status), the Pearson chi-square test was used.

A more complete analysis of our observational cohort study was used to identify our prediction models for conversion from HCs to MCI. In these models, we investigated taxonomic data (genera), functional data (GO), and clinical data (age, gender, BMI, and ApoE) as features separately, as well as in an ensemble learning model. Our aim was to find the most stable predictive model over time for the clinical outcome (MCI-converter), based on feature abundances. After normalization, our first aim was to reduce the number of included features. Following the approach by Brunner et al., a pre-selection of features was carried out by means of ANOVA-type statistics (ATS) [49]. The calculation was performed in R using the nparLD package (version 2.2) [50] with data from all three time instances. This approach resulted in the identification of an appropriate number of about 30 features per model using ATS, thus yielding longitudinal information. After suitable renormalization of the identified features, balances of the feature compositions at baseline were calculated and were used to train logistic regression models. Best baseline models for the different methods each (Genera, GO, and clinical meta data) were then applied to the data from 1yFU and 4yFU by means of a logistic regression approach. The discriminatory ability of the microbiome among both groups was investigated using receiver operating characteristic (ROC) analysis. In a last step, an ensemble learning model including the described models was investigated.

Customized R scripts and HeidiSQL (1.3) in connection with RMariaDB (1.1.1) were used for data analysis. R scripts relying on mlr (2.18.0) package were used for model training and feature selection [51]. For calculating the ROC curves, we employed OptimalCutpoints (1.1-4).

## Figures and Tables

**Figure 1 ijms-26-04735-f001:**
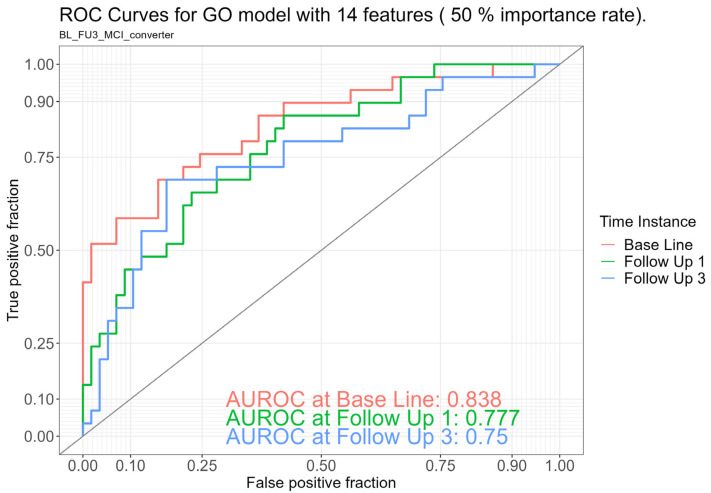
ROC curves for discrimination between MCI converters and stable healthy controls within a follow-up period of 4 years at baseline, 1-year follow-up (Follow Up 1), and 4-year follow-up (Follow Up 3) based on the Gene Ontology (GO) model including 14 features.

**Table 1 ijms-26-04735-t001:** Clinical and demographic characteristics of stable healthy controls (HCs) and converters from HC to mild cognitive impairment (MCI) after four years. * <0.05; ** <0.01.

	Stable HCs (*n* = 57)	HC-to-MCI Converters (*n* = 29)	*p*-Value
	**M (SD)**	**M (SD)**	
Age in years	71.8 (4.5)	71.6 (3.9)	0.79
MMSE—baseline	28.9 (2.3)	28.9 (1.1)	0.24
MMSE—1yFU	28.6 (1.0)	27.9 (1.5)	0.04 *
MMSE—4yFU	28.2 (1.6)	26.5 (1.1)	<0.001 **
GDS	1.8 (1.9)	2.2 (2.9)	0.49
BMI	25.6 (3.6)	26.6 (5.9)	0.44
	**Ratio (*n*:*n*)**	**Ratio (*n*:*n*)**	
Gender (male:female)	28:29	12:17	0.65
ApoE4 (e4 carriers:single e4 carriers:non-e4-carriers)	2:11:22	1:4:11	0.90
Arterial hypertension (yes:no)	25:32	12:17	0.83
Diabetes mellitus (yes:no)	3:54	1:28	0.71
Rheumatoid arthritis (yes:no)	2:55	2:27	0.60
NSAIDs (yes:no)	13:44	8:21	0.79
Anticoagulants (yes:no)	2:55	2:27	0.60
Antihypertensives (yes:no)	24:33	12:17	0.95
Antidiabetics (yes:no)	3:54	1:28	0.71
Statins (yes:no)	7:50	2:27	0.71
Antidepressants (yes:no)	1:56	3:26	0.11
AChE inhibitors (yes:no)	0:57	0:29	n.a.

Note: AChE = Acetylcholinesterase; ApoE = Apolipoprotein E; BMI = Body Mass Index; GDS = Geriatric Depression Scale; MMSE = Mini Mental State Examination; NSAIDs = Nonsteroidal antiphlogistics.

**Table 2 ijms-26-04735-t002:** Genus and phylum of the 38 features included in the genera model for discrimination between MCI converters and healthy persons. * <0.05; https://www.ncbi.nlm.nih.gov/Taxonomy/Browser/wwwtax.cgi?id=2 (accessed on 9 November 2023).

Genus	Phylum	Genera Levels in MCI Converters vs. Stable HCs (↑ Increased, ↓ Decreased)
Merdimonas	Bacillota/Firmicutes	↑ *
Butyricicoccus	Bacillota/Firmicutes	↑
Sharpea	Bacillota/Firmicutes	↓
Peptoanaerobacter	Bacillota/Firmicutes	↑
Brevundimonas	Pseudomonadota/a-proteobacteria	↑
Alkalibacter	Bacillota/Firmicutes	↑
Acetobacter	Pseudomonadota/aproteobacteria	↑
Phycicoccus	Actinomycetota	↑
Tepidanaerobacter	Bacillota/Firmicutes	↑
Natronincola	Bacillota/Firmicutes	↓
Anoxybacillus	Bacillota/Firmicutes	↓
Luteimonas	Pseudomonadota	↑
Azonexus	Pseudomonadota	↑
Gilvimarinus	Pseudomonadota	↑
Dehalococcoides	Chloroflexota	↑
Desulfovermiculus	Pseudomonadota	↓
Knoellia	Actinomycetota	↓
Roseisalinus	Pseudomonadota	↓
Polycyclovorans	Pseudomonadota	↑
Thiocystis	Pseudomonadota	↓
Sulfuricella	Pseudomonadota	↓
Methanococcoides	Euryarchaeota	↓
Oceaniovalibus	Pseudomonadota	↓
Numidum	Bacillota/Firmicutes	↓
Arcticibacter	Bacteroidota	↓
Agarivorans	Pseudomonadota	↓
Thermovenabulum	Bacillota/Firmicutes	↓ *
Herminiimonas	Pseudomonadota	↓
Natrinema	Euryarchaeota	↑
Oceanicoccus	Pseudomonadota	↓
Bergeriella	Pseudomonadota	↑
Endomicrobium	Elusimicrobiota	↓
Caviibacter	Fusobacteria	↑
Sagittula	Pseudomonadota	↓
Dehalobacter	Bacillota/Firmicutes	↓
Olegusella	Actinomycetota	↓
Labrenzia	Pseudomonadota	↑
Methanobrevibacter	Euryarchaeota	↓ *

**Table 3 ijms-26-04735-t003:** GO (Gene Ontology) labels, names, and pathways of 14 features included in the GO model for discrimination between MCI converters and healthy persons. * <0.05; ** <0.01; https://www.informatics.jax.org/vocab/gene_ontology/ and https://amigo.geneontology.org/ (accessed on 9 November 2023).

GO Label	Name/Term	Definition/Molecular Functions	Genera Levels in MCI Converters vs. Stable HCs (↑ Increased, ↓ Decreased)	
GO.0015416	ABC-type phosphonate transporter activity	Enables the transfer of a solute or solutes from one side of a membrane to the other according to the reaction: ATP + H_2_O + phosphonate(out) = ADP + phosphate + phosphonate(in).	↑ **	
GO.0051484	Isopentenyl diphosphate	Cholesterol pathway; mevalonate-independent pathway involved in terpenoid biosynthetic process.	↑ **	
GO.0004339	Glucan 1,4-alphaglucosidase activity	Enzyme in glycogenolysis: catalysis of the hydrolysis of terminal (1->4)-linked alpha-D glucose residues successively from non-reducing ends of the chains with release of beta-D-glucose.	↑ *	
GO.0046777	Protein autophosphorylation	The phosphorylation by a protein of one or more of its own amino acid residues (cis-autophosphorylation), or residues on an identical protein (trans-autophosphorylation).	↑	
GO.0009018	Sucrose phosphorylase activity	Catalysis of the reaction: sucrose + phosphate = D-fructose + alpha-D-glucose 1-phosphate.	↑	
GO.0043802	Hydrogenobyrinic acid a,c-diamide synthase (glutaminehydrolysing) activity, CopB	Part of the biosynthetic pathway to cobalamin in aerobic bacteria. Catalysis of the reaction: 2 L-glutamine + 2 ATP + 2 H_2_O + hydrogenobyrinate = 2 L-glutamate + 2 ADP + 4 H^+^ + hydrogenobyrinate a,c-diamide + 2 phosphate.	↑ *	
GO.0004415	Hyalurononglucosam inidase activity	Catalysis of the random hydrolysis of (1->4) linkages between N-Acetyl-beta-D-glucosamine and D-Dlucuronate residues in hyaluronate.	↑	
GO.0004631	Phosphomevalonate kinase activity	Catalysis of the reaction: (R)-5-phosphomevalonate + ATP = (R)-5-diphosphomevalonate + ADP + H^+^.	↑	
GO.0060567	Negative regulation of termination of DNA-templated transcription	Any process that decreases the rate, frequency, or extent of DNA-dependent transcription termination, the process in which transcription is completed.	↑ *	
GO.0004328	Formamidase activity	Catalysis of the reaction: formamide + H_2_O = formate + NH_4_.	↓	
GO.0018710	Acetone carboxylase activity	Catalysis of the reaction: acetone + ATP + CO_2_ + 2 H_2_O = acetoacetate + AMP + 4 H^+^ + 2 phosphate.	↑	
GO.0004491	Methylmalonate-semialdehyde dehydrogenase (acylating, NAD) activity	Synthesis of branched-chain amino acids, pyrimidine catabolic pathway catalysis of the reaction: 2-methyl-3-oxopropanoate + CoA + NAD+ = propanoyl-CoA + CO_2_ + NADH + H^+^.	↓ *	
GO.0016730	Oxidoreductase activity	Catalysis of an oxidation–reduction reaction in which an iron-sulfur protein acts as a hydrogen or electron donor and reduces a hydrogen or electron acceptor.	↓	
GO.0015667	DNA methyltransferase (cytosine-N4-specific) activity	DNA-modification, catalysis of the reaction: S-adenosyl-L-methionine + DNA cytosine = S-adenosyl-L-homocysteine + DNA N4-methylcytosine.	↓ **	

**Table 4 ijms-26-04735-t004:** Gut microbiome models predicting the progression from normal cognition to mild cognitive impairment (confidence intervals in brackets).

Models	Included Features	AUROC		
		Baseline (CI, %)	1yFU (CI, %)	4yFU (CI, %)
Taxonomic (Genera)	38	72% (60–84)	70% (59–82)	58% (45–71)
Functional (Gene Ontology [GO])	14	84% (75–93)	78% (68–88)	75% (63–87)
Clinical characteristics (Age, BMI, gender, and ApoE4)	4	57% (44–71)	54% (40–68)	55% (41–68)
Ensemble learning model		84% (75–93)	78% (68–88)	75% (63–87)

## Data Availability

The data sets of the present study are available at https://www.ebi.ac.uk/ena (available starting from 31 July 2025), PRJEB85717.

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
