# Peer review of "Prognostic Value of a Multivariate Gut Microbiome Model for Progression from Normal Cognition to Mild Cognitive Impairment Within 4 Years"

_ijms, 2025, doi:10.3390/ijms26104735_

Round 1

Reviewer 1 Report

Comments and Suggestions for Authors

The introduction could include more new publications in order to provide a more complete context for the research, as some references are too old and do not reflect the current state of research in this field.

The methodology section could be expanded by informing about the criteria for choosing features, or the reason for choosing the models used. the results section could be expanded by presenting more information on the context of the results of statistical research. For example, while it informs about the AUROC value for different models, it does not explain what these values mean in practical application for the prediction of MCI. In addition, the authors could add information about how risk factors such as age, gender, ApoE4 status can affect results.

The discussion section could present more information on the study's limitations. In addition, the discussion could be enriched by more detailed description of the results and their implications for clinical application, particularly how the analysis of the gut microbiome could be used in existing diagnostic frameworks.

The conclusion could focus more on the importance of the research in the field of neurodegenerative diseases in general, which would allow the reader to better understand the relevance of the study.

Author Response

Reply letter

Dear reviewer,

Thank you very much for your detailed feedback on our manuscript. We addressed your questions and hope to answer them sufficiently. Changes are colour-coded in the manuscript. If you have any further questions, please do not hesitate to contact us.

Reviewer 1 – changes coded in blue:

  1. The introduction could include more new publications in order to provide a more complete context for the research, as some references are too old and do not reflect the current state of research in this field.

Thank you for your important feedback. To address the current state of the research we included  more recent publication, e.g. two recent meta analyses focussing on the gut microbiome alterations in MCI as well a more recent publication on MCI. (p.2-3). If you have any further suggestions, please let us know.

“5-15% of patients with MCI develop Alzheimer’s disease (AD) per year. Therefore, MCI is often considered a precursor stage of developing dementia, such as AD (2).”

“Altered gut microbiome compositions are well documented in cross-sectional studies in patients with AD (7-11), and even in patients with preclinical AD (12). Furthermore, findings from recent meta-analyses confirm gut microbiological abnormalties in MCI and highlight their role as potential biomarkers for early identification and diagnosis of AD.  However, to assess prognostic validity of gut microbiome composition, longitudinal studies are needed.”

  1. The methodology section could be expanded by informing about the criteria for choosing features, or the reason for choosing the models used.

Thank you for pointing this out. For a better understanding, please find attached a more detailed description of the feature selection process:

For this purpose, two consecutive steps were carried out for feature selection:

Pre-selection of the features was conducted using the ANOVA-like test developed by Brunner, Bathke and Konietschke with the H0-hypothesis “no longitudinal development”. Significant features were selected, which resulted in a reduced, but still large number of features.

Thus, following the pre-selection process described in step 1, a further feature reduction was obtained by using repeated stepwise analysis of N logistic regression models, in which in each step, one variable was discarded to obtain the best model (N is the number of variables being preselected in step 1). This selection process was repeated about 1000 times.

For each of this N-1 models, data was randomly parted in five portions. Then, the first portion served as an internal validation data set, the remaining data were used for model training. The model was assessed using the internal validation data to calculate the accuracy. This was repeated five times (with each of the five portions used as internal validation set). Next the data were randomly parted to another set of five portions, and the whole training/validation cycle was repeated. This was done 30 times for each of the N-1 models, so that in the end a mean accuracy was obtained and the best model of this N-1 regressoin could be identified.

At the end, this resulted in about 1000 best models and their respective selected features. Based on the frequency of the selected features in these models, the most relevant features were identified to be incorporated in the final model. These models were trained with data from baseline and assessed with data from all time instances.

We hope that we were able to specify our selection process based on this specifictaion. However, after discussion in our team, we decided to include only few more information about the statistical selection process in the methods section (page. 15-16) in order to keep the statistical analyses description as simple and easily understandable as possible in the manuscript. Please let us know, if you wish us to specify the section even more.

  1. The results section could be expanded by presenting more information on the context of the results of statistical research. For example, while it informs about the AUROC value for different models, it does not explain what these values mean in practical application for the prediction of MCI.

This is a very important point. We addressed this issue in the introduction, when presenting our research question as follows (p. 3):

“This raises the question of whether such models could be informative for identifying healthy individuals who may later progress to MCI at an earlier disease stage, based on alterations in gut microbiome composition.”

Furthermore, we addressed your request in the discussion section (p. 8):

“This means e.g., that persons with MCI could be correctly identified with an accuracy of 75% after four years by using the GO model trained with microbiome data from baseline.”

And p. 12:

„[This evidence could inspire future studies focusing on the] specific clinical implications of our results (e.g. identification of specific gut microbiome profile alterations in healthy controls at risk for MCI; potential early interventions targeting the gut microbiome to reduce cognitive decline). “

  1. In addition, the authors could add information about how risk factors such as age, gender, ApoE4 status can affect results.

Thanks for this comment. We agree that these factors are important confounding variables in dementia research. For that reason we controlled for significant group differences concerning these factors. As no significant group differences between MCI and stable HC concerning these well-known risk-factors were found, we assume no confounding influences of these variables in our results.

  1. The discussion section could present more information on the study's limitations.

We thank the reviewer for this important comment. We included further study limitations also concerning our statistical approach in the discussion section (ps. 11-12)

“Despite the promising results, a few limitations of the study need to be addressed. One of our main limitations is our exploratory study design and the lack of a priori power calculation, which is especially critical concerning the large amount of possible features for statistical selection. Therefore, larger, more diverse, multi-center studies are needed to validate these findings in an independent sample, thereby increasing statistical power. Additionally, it is important to note that the analyzed models were only compared descriptively rather than statistically, meaning the superiority of the identified model is not definitively established. Furthermore, as we conducted a longitudinal observational study, we had to deal with the challenge of an imbalanced sample (more stable healthy controls compared to MCI converter), which might have influenced our results. As there are more complex statistical methods for variable selection (like e.g., LASSO, Boruta), future studies should include these statistical options. However, ANOVA type variable selection is a simple, but however valid option which furthermore allows us to compare our results with our former study. Additionally, the use of AUROCs in the statistical analyses does not distinguish between high sensitivity and specificity, which could be important for clinical applications.

Furthermore, contrary to our expectations, the clinical model including well established risk factors (age, gender, BMI, ApoE4) yielded only a poor prognostic value at all time points (AUROC ranging from 0.55 to 0.57), compared to the results of our previous study (13). This might be attributed to different sample characteristics, as these risk factors might be more relevant in the prediction of AD compared to MCI. This on the other hand strengthen our result of the predictive value of gut microbiome model in this earlier stage of disease. However, this might also question our statistical approach and the validity of our results and should therefore be incorporated in future studies. In comparison to our previous work, the different sample populations (conversion HC to MCI vs. MCI to AD) may explain the variations in models identified in this analysis. Future studies should focus also investigate the conversion from HC to AD. Last, it is important to note, that the reults of our analyses postulate a direct pathway between microbiome changes and cognitive decline. However, there might also be other relevant factors mediating this association like e.g. depression, cardiovasulare disease, metabolic disorders (Ng et al., 2023). “

  1. In addition, the discussion could be enriched by more detailed description of the results and their implications for clinical application, particularly how the analysis of the gut microbiome could be used in existing diagnostic frameworks.

We hope that we were already able to address this issue (see 3.). If you have any further requests concerning this recommendation, please do not hesitate to contact us again.

  1. The conclusion could focus more on the importance of the research in the field of neurodegenerative diseases in general, which would allow the reader to better understand the relevance of the study.

Thank you very much, we edited the conclusion section accordingly to your suggestion (p. 13).

“In conclusion, we identified stable gut microbiome algorithms that predict the progression from HC to MCI over a 4-year follow-up. The present findings highlight the prognostic value of the gut microbiome for the early identification of patients at increased risk for MCI. To our knowledge, this is the first study investigating gut microbiome features in healthy individuals as prognostic markers for MCI, confirming a link between gut microbiome dysbiosis and cognitive decline. So far, previous research established the composition of the gut microbiome as a potential contributing factor of the development of neurodegenerative diseases, such as AD. By focusing specifically on the transitional stage of MCI, our research extends the field of neurodegenerative disease studies at a much earlier point in the disease’s progression and complement the results of our previous study. Our findings might be informative for designing future studies focussing on concrete diagnostic tools in clinical practice, potentially enhancing early detection of these conditions by using gut microbiome data. If gut microbiome analysis proves effective as a diagnostic tool, it could enable early preventive non-invasive, or therapeutic interventions. Notably, microbiome signatures may be detectable in cognitively asymptomatic individuals, potentially indicating future MCI progression. Especially relevant to clinical practice might be the identification of gut microbiome signatures that are unique to MCI with an underlying AD compared to MCI due to other etiologic causes. Thus, research in that field could overall provide a critical time advantage for early diagnosis and targeted interventions. Future studies should build on these findings to explore the causal relationship between microbiome dysbiosis and MCI development, ultimately improving diagnostic accuracy and potentially leading to the development of early interventions for MCI, preclinical dementia, and other neurodegenerative diseases.”

Reviewer 2 Report

Comments and Suggestions for Authors

The manuscript is interesting and enrich the advancement regarding relationship of microbiota and (physiological or pathological) neuron conditions, as well as the conditions in healthy patients linked to the  early development of cognitive deficit. 

Some changes are suggested:

a) The abstract should be specific regarding the findings of microbiota related as predictors of MCI.

b) Introduction should be enriched with scope in the field.

c) The discussion should be clear regarding limitations and data which could bias the interpretation of results.

d) The content of Figure 1 and Table 4. should be described in results and(or) clearly linked to discussion.

e) Conclusions should be concise, but clear in the innovation comparatively to your previous work. Also the inclusion of some limitation is desirable before the sentence starting with "future studies should..."

Author Response

Reply letter

Dear reviewer,

Thank you very much for your detailed feedback on our manuscript. We addressed your questions and hope to answer them sufficiently. Changes are colour-coded in the manuscript. If you have any further questions, please do not hesitate to contact us.

Reviewer 2 – changes coded in yellow

1) The abstract should be specific regarding the findings of microbiota related as predictors of MCI.

Thank you so much for this very important point. We changed the abstract as follows (p. 1):

“Little is known about the dysbiosis of the gut microbiome in patients with mild cognitive impairment (MCI) potentially at risk for the development of Alzheimer’s disease, (AD).  So far, only cross-sectional differences and not longitudinal changes and their prognostic significance has been in the scope of research in MCI.“

2) + 3) + 6) Introduction should be enriched with scope in the field. & The discussion should be clear regarding limitations and data which could bias the interpretation of results. Also the inclusion of some limitation is desirable before the sentence starting with "future studies should..."

Thank you very much. As these issues were also addressed by reviewer 1, we added some newer references and meta-analysis in the introduction (see blue-coded changes on ps. 2-3) and limitations in the discussion section (blue-coded changes on ps. 11-12).

4) The content of Figure 1 and Table 4. should be described in results and(or) clearly linked to discussion.

Thank you for this informative point. We checked our manuscript for the above mentioned references. We referred to Figure 1 in the result section (p. 5). Table 4 is a summarizing overview of the results we presented separately in the results section, we added a short description (p. 4).

5) Conclusions should be concise, but clear in the innovation comparatively to your previous work.

Thank you very much, as again reviewer 1 also suggested this point, we edited the conclusion section accordingly (p. 13).

“In conclusion, we identified stable gut microbiome algorithms that predict the progression from HC to MCI over a 4-year follow-up. The present findings highlight the prognostic value of the gut microbiome for the early identification of patients at increased risk for MCI. To our knowledge, this is the first study investigating gut microbiome features in healthy individuals as prognostic markers for MCI, confirming a link between gut microbiome dysbiosis and cognitive decline. So far, previous research established the composition of the gut microbiome as a potential contributing factor of the development of neurodegenerative diseases, such as AD. By focusing specifically on the transitional stage of MCI, our research extends the field of neurodegenerative disease studies at a much earlier point in the disease’s progression and complement the results of our previous study. Our findings might be informative for designing future studies focussing on concrete diagnostic tools in clinical practice, potentially enhancing early detection of these conditions by using gut microbiome data. If gut microbiome analysis proves effective as a diagnostic tool, it could enable early preventive non-invasive, or therapeutic interventions. Notably, microbiome signatures may be detectable in cognitively asymptomatic individuals, potentially indicating future MCI progression. Especially relevant to clinical practice might be the identification of gut microbiome signatures that are unique to MCI with an underlying AD compared to MCI due to other etiologic causes. Thus, research in that field could overall provide a critical time advantage for early diagnosis and targeted interventions. Future studies should build on these findings to explore the causal relationship between microbiome dysbiosis and MCI development, ultimately improving diagnostic accuracy and potentially leading to the development of early interventions for MCI, preclinical dementia, and other neurodegenerative diseases.”

Reviewer 3 Report

Comments and Suggestions for Authors
  1. The study title should indicate the study design.
  2. The introduction should introduce some of the key research questions/hypotheses the study seeks to investigate.
  3. The findings should be reported according to TRIPOD-AI guidelines and a copy of the completed checklist should be appended for review.
  4. How was the sample size determined? The dataset is too small relative to the number of microbial features considered. Microbiome data is high-dimensional, with only 29 MCI converters, the number of positive cases is far below the number of features. Machine learning models typically require at least 10 to 15 events per feature for stable training.
  5. Given the imbalances in study converter vs. non-converter groups (29 converters, 57 stable controls), logistic regression may be biased toward the majority class. The AUROC metric is also misleading in imbalanced datasets. As such, precision-recall curves should be reported.
  6. How was missing data handled? At least some comments are necessary.
  7. Further explaining and justification is required for the features selection process. Feature selection should ideally be data-driven (e.g., LASSO, Boruta, or SHAP-based feature importance), rather than an arbitrary statistical cutoff (e.g., ANOVA-type filtering).
  8. The gut microbiome has thousands of features, yet the authors provide only vague descriptions of which microbial taxa and functional pathways were predictive. Table 2 and 3 list some genera and GO features, but why were these chosen? More explanation and justification are required.
  9. No confusion matrices or calibration curves are provided to assess model reliability.
  10. I am still unsure why specific functional features (e.g., ABC transporters) should be considered better predictors than others. 
  11. The discussion section needs to be more critical. If the clinical model (AUROC ~0.55) was weak, then is the microbiome model truly adding predictive power, or is it an artifact of overfitting?
  12. I tend to disagree with the authors' claim that the functional model remained "stable" over time. Notably, the model's AUROC dropped from 0.84 at baseline to 0.75 at 4-year follow-up. This suggests some level of degradation in predictive accuracy.
  13. MCI is not a stable entity; some cases revert to normal cognition. No details are provided about how MCI was diagnosed beyond MMSE, which is unable to distinguish true cognitive decline from normal aging.
  14. The study attributes differences in gut microbiome to MCI progression, but alternative explanations are possible. There are also other age-related conditions (e.g., depression, cardiovascular disease, metabolic disorders) that could drive both microbiome changes (citation: pubmed.ncbi.nlm.nih.gov/36986088) and cognitive decline.
  15. For the IRB approval statement, please also include the date of approval.
Comments on the Quality of English Language

Moderate edits required.

Author Response

Reply letter

Dear reviewer,

Thank you very much for your detailed feedback on our manuscript. We addressed your questions and hope to answer them sufficiently. Changes are colour-coded in the manuscript. If you have any further questions, please do not hesitate to contact us.

Reviewer 3 – coded in grey

1) The study title should indicate the study design.

Thank you very much for your suggestion. Following the TRIPOD-AI guideline, we changed the title to “Prognostic value of a multivariate gut microbiome model for progression from normal cognition to mild cognitive impairment within 4 years”. We hope to meet your expectations. If not, please specify what you would like us to include in the title.

2) The introduction should introduce some of the key research questions/hypotheses the study seeks to investigate.

Thank you for your recommendation. As the research field is still growing and so far only cross-sectional studies are available which were not able to identify one specific genus or functional pathway, our study is of exploratory nature. Therefore, we did not have concrete hypotheses e.g. which specific genera might predict changes in cognitive decline. For that reason, we already included our main research question in our first draft. Following your suggestion and also reviewer 1’s suggestion, we specified the main research question as follows (p. 3):

“This raises the question of whether the development of such prediction models could also be informative at an earlier stage of disease. Therefore our main research question was to identify the most stable multivariate model prediciting the progression from normal cognition to MCI based on gut microbiome composition.“

3) The findings should be reported according to TRIPOD-AI guidelines and a copy of the completed checklist should be appended for review.

Thank you very much, we included the checklist in the reply letter and added the missing points. However, some points are not applicable to our study design. As our study is the first in this field, to investigate the predicitve validity of gut mircobiome in the predition of conversion from HC to MCI, our study results are of exploratory nature and do not have the goal to result in a prediction model being applicable to other populations. Our goal was to investigate the potentail causal link between gut microbiome and cognitive devline by being the first to use a longitudinal design, while so far, only cross-sectional data has been available. 

4) How was the sample size determined? The dataset is too small relative to the number of microbial features considered. Microbiome data is high-dimensional, with only 29 MCI converters, the number of positive cases is far below the number of features. Machine learning models typically require at least 10 to 15 events per feature for stable training.

Thank you very much, we agree that this is a very important statistical point. In this study, no power-analysis was made a-priori. To our knowledge, compared to other clinical prediction model studies in this field (e.g. Cerroni et al., 2022), our sample is still considerably large and in our opinion sufficient enough for a first exploratory study. Furthermore, following Brunner & Bathke (2018), for feature selection we trained the model only with baseline data and validated the results using all time instance, which might strengthen results found in our study (see also Analysis section, p. 14-15). However, we agree that larger sample sizes are needed to validate the exploratory results. Therefore, we included it as a limitation in our discussion section, see p. 27.

“One of our main limitations is our exploratory study design and the lack of a priori power calculation, considering the large amount of possible features for statistical selection.”

Furthermore, we specified it in our Methods section, p. 15.

“No a-priori power analysis was conducted to assess the sample size.”

5) Given the imbalances in study converter vs. non-converter groups (29 converters, 57 stable controls), logistic regression may be biased toward the majority class. The AUROC metric is also misleading in imbalanced datasets. As such, precision-recall curves should be reported.

Thank you very much for this important point. We fully agree with you on that topic that a balanced sample would be better for the conducted statistical analyses. As we did a longitudinal clinical observational study, however, controlling for a balanced sample is challenging as the rate of healthy participants developing MCI at 1 FU and 4 FU is expected to be smaller than the rate of healthy individuals remaining stable at the follow up session. As this is an observational study, we therefore have to deal with the challenge of an imbalanced sample. However, we assume that the imbalance in our groups lies within an acceptable range and allows for our statistical analyses.

Furthermore, we added the following sentence to the limitation section:

“Furthermore, as we conducted a longitudinal observational study, we had to deal with the challenge of an imbalanced sample (more stable healthy controls compared to MCI converter), which could also influence our results. “

6) How was missing data handled? At least some comments are necessary.

We conducted a completer analysis, therefore there was no need to handle missing data. Thank you for this important point, we included information in our analysis section (p. 15)

7) Further explaining and justification is required for the features selection process. Feature selection should ideally be data-driven (e.g., LASSO, Boruta, or SHAP-based feature importance), rather than an arbitrary statistical cutoff (e.g., ANOVA-type filtering).

We again thank the reviewer for this important point. We admit that compared to other more complex methods like LASSO, an ANOVA-type variable selection is a more simple technique, which however normally results in a very low Type II error and also Type I error, which makes ANOVA type filtering a suitable option for variable selection (e.g. Kirpich et al., 2018). Furthermore, one important point for us for selecting this type of analysis was, that we wanted to keep the statistical analyses comparabel to our former study (Laske et al., 2024) investigating the predictive validity of gut microbiome for the transition vom MCI to AD. However, we admit that future studies should also focus on data-driven, more complex options (like e.g., LASSO, Boruta, etc.).

We therefore added this point to the discussion section as limitation:

“As there are more complex statistical methods for variable selection (like e.g., LASSO, Boruta), future studies should include these statistical options. However, ANOVA type variable selection is a simple, but however valid option which allows us to compare our results with our former study.”

Reference: Kirpich A, Ainsworth EA, Wedow JM, Newman JRB, Michailidis G, McIntyre LM (2018) Variable selection in omics data: A practical evaluation of small sample sizes. PLoS ONE 13(6): e0197910. https://doi.org/10.1371/journal. pone.0197910

8) The gut microbiome has thousands of features, yet the authors provide only vague descriptions of which microbial taxa and functional pathways were predictive. Table 2 and 3 list some genera and GO features, but why were these chosen? More explanation and justification are required.

Thank you very much for this important comment. The genera and GO features listed in Table 2 and 3 are the results of our statistical selection process using ANOVA-type filtering, this is why only these are reported.

9) No confusion matrices or calibration curves are provided to assess model reliability.

Thank you very much, for addressing your concern, we included confidence intervals to assess model reliability (Table 4, p. 8)

10) I am still unsure why specific functional features (e.g., ABC transporters) should be considered better predictors than others.

Thank you very much for your question. As we mentioned above, the functional features in Table 3 are already the results of our statistical selection process using ANOVA-type statistic. Descriptively comparing the models, the functional model outperformed the genera-based model and was as good at prediction as the ensembled learning model, which leads us to the above-mentioned interpretation of our data. 

11) The discussion section needs to be more critical. If the clinical model (AUROC ~0.55) was weak, then is the microbiome model truly adding predictive power, or is it an artifact of overfitting?

Thank you very much for this very important point. We added a respective point in our discussion section, starting at p. 11.

“Furthermore, contrary to our expectations, the clinical model including well established risk factors (age, gender, BMI, ApoE4) yielded only a poor prognostic value at all time points (AUROC ranging from 0.55 to 0.57), compared to the results of our previous study (13). This might be attributed to different sample characteristics, as these risk factors might be more relevant in the prediction of AD compared to MCI. This on the other hand strengthen our result of the predictive value of gut microbiome model in this earlier stage of disease. However, this might also question our statistical approach and the validity of our results and should therefore be incorporated in future studies.”

12) I tend to disagree with the authors' claim that the functional model remained "stable" over time. Notably, the model's AUROC dropped from 0.84 at baseline to 0.75 at 4-year follow-up. This suggests some level of degradation in predictive accuracy.

Thank you very much, we reformulated the respective section (p.8) and specified that there is a decline from 0.84 to 0.75. Furthermore, we already incorporated this point in our discussion section on p. 9, l. 185 ([Additionally,] prognostiv power slightly decereased over time in the presen study [….].

13) MCI is not a stable entity; some cases revert to normal cognition. No details are provided about how MCI was diagnosed beyond MMSE, which is unable to distinguish true cognitive decline from normal aging.

Thank you very much for your comment. We specified the MCI diagnostic process in the Method section, ps. 13-14).

“The subjects participating in the AlzBiom study were recruited at the Section of Dementia Research at the Department of Psychiatry and Psychotherapy in Tübingen. All participants were clinically examined at BL by means of routine diagnostic work-up for dementia including physical, neurological and psychatric examinations as well as brain imaging. At BL, inclusion criteria were normal cognition 1) assessed via Mini-Mental State Examination ((37); MMSE ≥ 27) and the Clinical Dementia Rating scale (38, 39) CDR = 0), 2) no subjective cognitive decline (SCD), and 3) no history of neurological or psychiatric disorders. For MCI detection, we used the amnestic MCI criteria proposed by Petersen et al. (41). Furthermore, MMSE and CDR were repeated at 1yFU and 4yFU.“

14) The study attributes differences in gut microbiome to MCI progression, but alternative explanations are possible. There are also other age-related conditions (e.g., depression, cardiovascular disease, metabolic disorders) that could drive both microbiome changes (citation: pubmed.ncbi.nlm.nih.gov/36986088) and cognitive decline.

Thank you very much for your point, we totally agree with you on that matter.

 In our study, we tried to control for the most important factors by comparing the most important confounding factors at baseline (including e.g. GDS for depression screening; for more factors, we refer to Table 1). However, you are right, that these factors could also act as mediating variable, which was however not part of our analyses. We therefore added this as limitation in our discussion section as follows:

“It is important to note, that the reults of our analyses postulate a direct pathway between microbiome changes and cognitive decline. However, there might also be other relevant factors mediating this association like e.g. depression, cardiovasulare disease, metabolic disorders (Ng et al., 2023).” 

 15) For the IRB approval statement, please also include the date of approval.

Thank you, we included the necessary information, p.16.

“Institutional Review Board Statement: The study was conducted in accordance with the Declaration of Helsinki, and approved by the Ethics Committee of the Medical Faculty of Eberhard-Karls-University and University Hospital Tübingen (protocol code 721/2015BO2, 18.09.2019

Reviewer 4 Report

Comments and Suggestions for Authors

The authors evaluated taxonomic and functional models based on microbiome data in order to predict individuals converting from health to MCI over a time frame of 4 years. Overall, the introduction and methods sections are detailed, well-structured, and the research hypothesis is clearly outlined. The results are well presented and potential mechanisms are proposed. Actually, this paper can be published as the present form.

Only a few minor concerns:

  1. Keyword: Taxonomic data, functional data, longitudinal data are nonspecific scientific terms.
  2. Abstract: “offering a promising innovative supplement for early AD prognosis” Identification is more accurate than prognosis.
  3. Advise to draw spaghetti plots as their previous work, i.e., Laske et al., 2024 IJMS.
  4. Limitation: Dose any other neurodegenerative diseases can be examined by these model to predict cognition decline?

Author Response

Reply letter

Dear reviewer,

Thank you very much for your detailed feedback on our manuscript. We addressed your questions and hope to answer them sufficiently. Changes are colour-coded in the manuscript. If you have any further questions, please do not hesitate to contact us.

Reviewer 4 – coded in green

1) Keyword: Taxonomic data, functional data, longitudinal data are nonspecific scientific terms.

Thank you very much for your suggestion, we changed the key words following your advice.

“Alzheimer’s disease; mild cognitive impairment, longitudinal observational study, prediction model, gut microbiome”

2) Abstract: “offering a promising innovative supplement for early AD prognosis” Identification is more accurate than prognosis.

Thank you very much, we changed the abstract accordingly (p. 1)

“Thus, gut microbiome algorithms have the potential to predict MCI conversion in HCs over a 4-year period, offering a promising innovative supplement for early AD identification.”

3) Advise to draw spaghetti plots as their previous work, i.e., Laske et al., 2024 IJMS.

Thank you very much for your suggestion and thank you for finding the spaghetti plots in our previous work informative. In this work, we aimed to keep the manuscript as concise as possible, hence refrained from spaghetti plots this time, as we did not want to focus on specific genera in the presentation of the features.

4) Limitation: Dose any other neurodegenerative diseases can be examined by these models to predict cognition decline?

Thank you very much for your question. In this work, we mainly aimed to investigate gut microbiome composition changes that potentially are linked to the progression to MCI. In the overall study, we were interested in the progression of patients transitioning to Alzheimer’s disease, hence did not specifically investigate other neurodegenerative diseases. However, these approaches have also been used in other neurodegenerative disorders (like e.g. Parkinson, see p.8).

Round 2

Reviewer 1 Report

Comments and Suggestions for Authors

I consider now the article proper to be accepted in the journal, the authors made the necessary corrections and adding information

Author Response

Dear reviewer, thank you very much!

Reviewer 3 Report

Comments and Suggestions for Authors

Thank you for the replies and revisions. Some remaining pointers:

1. Although the authors acknowledge exploratory nature of their findings and the fact that there were no a priori power calculations, this is still a major issue for a machine learning study with high-dimensional features (especially in omics). With only 86 participants and 29 converters, overfitting is a serious risk, particularly with 38 genera or 14 GO terms used as predictors. The authors should perform power simulations or use bootstrapping to estimate variance and validate model performance more robustly.

2. Using ANOVA-type statistics for feature selection is a limitation and this should be duly acknowledged.

3. The manuscript compares gut microbiome models with a questionably weak clinical model (age, gender, BMI, ApoE4 only). This is a straw man comparator, and not reflective of real-world clinical models that might include neuroimaging, CSF biomarkers, or neuropsychological tests. Suggest to temper the conclusions and clarify that the goal is not to replace but to supplement existing clinical models.

Comments on the Quality of English Language

Minor edits only.

Author Response

Dear reviewer,

again, thank you very much for your feedback. We edited parts of the manuscripts following your suggestions which are again colour-coded in grey in the file.

Regarding the remaining pointers:

  1. Although the authors acknowledge exploratory nature of their findings and the fact that there were no a priori power calculations, this is still a major issue for a machine learning study with high-dimensional features (especially in omics). With only 86 participants and 29 converters, overfitting is a serious risk, particularly with 38 genera or 14 GO terms used as predictors. The authors should perform power simulations or use bootstrapping to estimate variance and validate model performance more robustly.

Thank you again for your important comment. We understand your major concern regarding the serious risk of overfitting in our analyses.

First, we totally agree that this should be more clearly highlighted in the discussion section. Therefore, we added the following specification at the beginning of our limitation paragraph:

“One of our main limitations is our exploratory study design and the lack of a priori power calculation, which is especially critical concerning the large number of possible features for statistical selection which might result in a serious risk of overfitting.”

Taking a closer look at the confindence intervalls of our models, there is an evident range in these values. Nevertheless, regarding our best performing model (GO model), the CI still ranges clearly above random chance over all three time points, thus supporting our identified model.  

Second, the use of bootstrapping is highly indicated in this context. For this reason, we would like to clarify that bootstrapping is already included in our conducted analyses. We admit that this was not clearly described in our methods section and was indeed also mentioned by another reviewer. So far, we tried to keep our methods section as easily understandable and concise as possible, which is why we refrained from giving a detailed description of the different steps in our analyses. For a better understanding, please find attached more details about the analysis process.

For this purpose, two consecutive steps were carried out for feature selection:

Pre-selection of the features was conducted using the ANOVA-like test developed by Brunner, Bathke and Konietschke with the H0-hypothesis “no longitudinal development”. Significant features were selected, which resulted in a reduced, but still large number of features.

Thus, following the pre-selection process described in step 1, a further feature reduction was obtained by using repeated stepwise analysis of N logistic regression models, in which in each step, one variable was discarded to obtain the best model (N is the number of variables being preselected in step 1). This selection process was repeated about 1000 times.

For each of this N-1 models, data was randomly parted in five portions. Then, the first portion served as an internal validation data set, the remaining data were used for model training as it is done in bootstrapping. We used function implemented in the mlr package (cf. [1]) in combination with the balance approach featured in [2]. The model was assessed using the internal validation data to calculate the accuracy. This was repeated five times (with each of the five portions used as internal validation set). Next the data were randomly parted to another set of five portions, and the whole training/validation cycle was repeated. This was done 30 times for each of the N-1 models, so that in the end a mean accuracy was obtained and the best model of this N-1 regression could be identified.

At the end, about 18 Mio. regression steps resulted in about 1000 best models and their respective selected features. Based on the frequency of the selected features in these models, the most relevant features were identified to be incorporated in the final model. These models were trained with data from baseline and assessed with data from all time instances.

We hope that with our response we were able to address the above-mentioned points, especially by clarifying that our statistical procedure included bootstrapping. In case there are still open questions from your side, please let us know.

[1] Hefin I. Rhys: Machine Learning with R, the tidyverse, and mlr. Shelter Island (NY) 2020

[2] Rivera-Pinto JEgozcue JJ, Pawlowsky-Glahn V, Paredes RNoguera-Julian M, Calle ML. 2018. Balances: a New Perspective for Microbiome Analysis. mSystems 3:10.1128/msystems.00053-18. https://doi.org/10.1128/msystems.00053-18

  1. Using ANOVA-type statistics for feature selection is a limitation and this should be duly acknowledged.

                We thank the reviewer for this comment and critically reformulated the use of ANOVA-type statistics in our limitation section:

“Regarding our statistical methods in more detail, ANVOA type statistics for feature selection might be a valid, but rather arbitrary statistical option with several limitations. Therefore, future studies should use more complex, data-driven statistical methods  like e.g., LASSO, Boruta instead.”

  1. The manuscript compares gut microbiome models with a questionably weak clinical model (age, gender, BMI, ApoE4 only). This is a straw man comparator, and not reflective of real-world clinical models that might include neuroimaging, CSF biomarkers, or neuropsychological tests. Suggest to temper the conclusions and clarify that the goal is not to replace but to supplement existing clinical models.

Thank you again for your comment. We totally agree with you that in routine clinical practice the above-mentioned clinical features would also be included in the diagnosis of dementia which might, at first glance, question the validity of our clinical model. However, in earlier stages of disease, the above-mentioned clinical markers frequently remain inconclusive and are partly of invasive nature. Gut microbiome markers, on the other hand, might be an especially useful, easily collectable diagnostic supplement at this stage of disease. However, we do understand your concern and indeed see the usefulness of gut microbiome as a diagnostic supplement and not replacement of well-established diagnostic assessments. Therefore, we adapted the conclusion section as follows:

“In conclusion, we identified stable gut microbiome algorithms that predict the progression from HC to MCI over a 4-year follow-up. The present findings highlight the prognostic value of the gut microbiome for the early identification of patients at increased risk for MCI, which might supplement existing diagnostic assessment methods. […..] Future studies should build on these findings to explore the causal relationship between microbiome dysbiosis and MCI development, in order to improve diagnostic accuracy and supplement existing diagnostic assessment methods, thereby facilitating the development of early interventions for MCI, preclinical dementia, and other neurodegenerative diseases.”

Please contact us for any further questions.